# Functional Characterization of Ammonium Transporter MhAMT1;2 in *Malus hupehensis*

**Jiazhen Li, Muting Yu, Huibin Li, Guangkai Yang, Linlin Huang * and Yanyan Hao ***

College of Horticulture, Shanxi Agricultural University, Jinzhong 030801, China
* Correspondence: llhuang@sxau.edu.cn (L.H.); yanyanhao123@163.com (Y.H.)

**Abstract:** The absorption and utilization of $NH_4^+$ and $NO_3^-$ by plant roots is closely related to soil moisture. In this study, we investigated the effect of short-time drought and rewatering on uptake and assimilation of $NH_4^+$ and $NO_3^-$ in 1-year-old *Malus hupehensis* plants, as well as transcription changes of *ammonium transporters* (*AMTs*) and *nitrate transporters* (*NRTs*). In roots, the $NH_4^+$ and $NO_3^-$ content and nitrate reductase activity decreased under drought and to some extent was restored after rewatering. Expression analysis indicated that most investigated *AMTs* and *NRTs* were down-regulated while *MhAMT1;2* was significantly up-regulated in drought-stressed roots. Therefore, the function of *MhAMT1;2* was further studied through bioinformatics analysis, tissue-specific expression, subcellular localization, and functional complementation in $NH_4^+$ uptake-defective yeast and Arabidopsis mutants. Results showed that *MhAMT1;2* was mainly expressed in roots and localized to the cell membrane. Moreover, MhAMT1;2 can mediate $NH_4^+$ uptake in both yeast and Arabidopsis mutants, and the transport process was affected by external proton concentrations and ATP. The present study will create a basis for exploring the functional roles of plant *AMT* members and improving N uptake and use efficiency under drought condition in fruit trees.

**Keywords:** ammonium transporter; function analysis; drought; nitrogen uptake





## 1. Introduction

Nitrogen (N) is one of the most abundant elements required for plant growth and yield. Absorption of inorganic N from soil is a primary way for plants to obtain N sources [1]. In most agricultural ecosystems, ammonium ($NH_4^+$) and nitrate ($NO_3^-$) are the main inorganic N sources in the soil [2]. The uptake of $NH_4^+$ and $NO_3^-$ by plants is closely related to the state of soil moisture, as they are firstly dissolved in water and subsequently absorbed by plant roots. Therefore, many previous studies are concerned with the uptake and utilization efficiency of inorganic N sources under long-term drought conditions [3–5]. Indeed, more research is still needed to explore the changes of $NH_4^+$ and $NO_3^-$ absorption by plants under progressive water deficit and rewatering.

Previous studies have detected reduced absorption of $NH_4^+$ and/or $NO_3^-$ in many plant species suffering from prolonged drought stress [3,6], perhaps due to less energy at the whole plant level. Noteworthily, relatively enhanced uptake of $NH_4^+$ has been reported in roots of drought-stressed plants when compared with $NO_3^-$ [7,8]. The $NH_4^+$ concentration also increased significantly in the roots of *Sophora japonica* under drought conditions [9]. Improving the absorption capacity of plant roots to $NH_4^+$ can not only promote N use efficiency [10], but can also be related to plant drought tolerance. Studies in *Populus simonii* have shown that increased N nutrition improved plant drought tolerance by increasing $NH_4^+$ uptake and reducing N metabolism [7]. $NH_4^+$ supply promoted the root growth and root hydraulic conductivity of rice (*Oryza sativa*) under drought stress, so as to improve its drought resistance [11]. Furthermore, $NH_4^+$ also triggered the antioxidant cellular machinery to protect citrus plants from osmotic stress [12]. Considering the close relationship between improving $NH_4^+$ absorption and drought tolerance, the way in which a plant absorbs and utilizes $NH_4^+$ to protect it from drought deserves further study.

Plants take up and transport $NH_4^+$ mainly through members of the $NH_4^+$ transporter (AMT) family [7,12]. By now, the AMT gene family has been reported in many plant species such as *Arabidopsis thaliana* [13], *Solanum lycopersicum* [14], *Lotus japonicus* [15], and *O. sativa* [16]. Particularly, several key *AMT* genes have been functionally studied and identified. For instance, the triple mutants of *OsAMT1;1*, *OsAMT1;2* and *OsAMT1;3* resulted in a decrement in plant growth and yield by reducing 90–95% of $NH_4^+$ absorption [17], and overexpression of *OsAMT1;1* led to superior growth and higher yield [18]. *PsAMT1.2*-overexpressing transgenic poplar plants exhibited higher salt resistance with increased photosynthetic rate and plant height compared with wild types [19]. *PutAMT1;1* from *Puccinelia tenuiflora* also enhanced plant salt tolerance in Arabidopsis at the early stage of root growth when being overexpressed [20]. As reported in previous studies, the transcript level of certain *AMTs* altered obviously in roots of drought-stressed plants, some of which were prominently up-regulated, e.g., *TaAMT1.2* in *Triticum aestivum* [21], *PsAMT1;2/1;6* in *P. simonii* [7] and *SjAMT1;1* and *SjAMT2;1a* in *S. japonica* [9]. These valuable genes mentioned above are worthy of further study to figure out their roles in $NH_4^+$ uptake during plant adaption or resistance to drought stress.

*Malus hupehensis* Rehd. is a kind of widely used apple rootstock in China, and its advantage is high genetic stability by reasons of apomixis. Our previous study has found that $NH_4^+$ absorption by roots of *M. hupehensis* was significantly promoted under PEG-induced drought stress, which is further confirmed by non-invasive micro-test (NMT) and [15]N stable isotope tracer [7]. In addition, we have identified 15 *AMT* family members in the apple genome [22]. In the present study, we conducted a short-time drought and rewatering treatment. We measured the changes in the concentrations of $NH_4^+$ and $NO_3^-$, activities of enzymes involved in N assimilation, and expression levels of *AMTs* and $NO_3^-$ transporters (*NRTs*) in roots and leaves of one-year old *M. hupehensis* plants. Additionally, we further studied the function of *MhAMT1;2*, which was up-regulated under drought stress, by means of bioinformatics analysis, subcellular localization, and functional complementation in $NH_4^+$ uptake-defective yeast and Arabidopsis.

## 2. Materials and Methods

### 2.1. Plant Material and Treatment

In this study, one-year-old potted plants of *M. hupehensis* were used as materials and cultured in a greenhouse (25–28 °C/15–18 °C day/night). A selection of 30 plants of similar size (about 110 cm tall) were watered with saturated water one day in advance, then treated for 7 days without any water, and rewatered on the evening of 7th day. The mature leaves and new roots of 6 random plants were sampled on 9:00 at 1, 3, 5, 7 and 8 d after saturated water, and the leaf relative water content was measured at the same time. For tissue-specific expression analysis, the root, stem, young leaf, mature leaf and senescing leaf of the *M. hupehensis* plants with normal irrigation were collected at 9:00, and rapidly frozen in liquid N before being stored at −80 °C.

### 2.2. Determination of $NO_3^-$, $NH_4^+$ and Activities of Enzymes Involved in N Assimilation

Determination of $NO_3^-$ content was performed as described by Huang et al. [23]. $NH_4^+$ content was determined according to the Berthelot reaction [24]. Nitrate reductase (NR) activity was determined as previously described [25]. Glutamine synthesis (GS) activity was performed as described by Yu et al. [26].

### 2.3. Expression Analysis of AMTs

Total RNA was isolated using Plant Total RNA Isolation Kit (Foregene, Chengdu, China) and reverse transcribed using a HiScript II 1st Strand cDNA Synthesis Kit with gDNA wiper (Vazyme, Nanjing, China). Specific primers (Table S1) for *AMTs* and *NRTs* were designed to perform qRT-qPCR using the LightCycler 480 real-time PCR system (Roche, Basel, Switzerland) with ChamQ Universal SYBR qPCR Master Mix (Vazyme, Nanjing, China). The PCR condition for thermal cycling was as follows: 95 °C for 30 s,

40 cycles of 95 °C for 5 s and 60 °C for 30 s. The relative expression of *AMTs* was calculated against the expression levels of the housekeeping gene ACTIN using the $2^{-\Delta\Delta CT}$ method, where the cycle threshold (CT) values were obtained from three independent biological replicates [27].

### 2.4. Generation of MhAMT1;2 cDNA

Total RNA from the roots of *M. hupehensis* was used to reverse transcribe into cDNA with HiScript II 1st Strand cDNA Synthesis Kit with gDNA wiper (Vazyme, NanJing, China). The open reading frame (ORF) was generated by high fidelity PCR amplification using the specific primers (Table S1). The resulting products were ligated to pMD18-T vector (Takara, Beijing, China), and transformed into *Escherichia coli* Trans5α (ANGYUBIO, Shanghai, China). The colonies were verified by PCR and agarose gel electrophoresis, and the positive clones producing 1500 bp bands were sequenced (Sangon Biotech, Shanghai, China).

### 2.5. Bioinformatics Analysis of MhAMT1;2

The biological online software TMHMM Server, v.2.0 (https://services.healthtech.dtu.dk/service.php?TMHMM-2.0, accessed on 13 January 2023) was used to predict the protein transmembrane domain. The protein sequences in full length for *A. thaliana*, *O. sativa*, *Populus trichocarpa*, *Sorghum bicolor*, *L. esculentum* and *Lotus japonicas* were downloaded from the NCBI protein database to reveal their evolutionary relationship. Full-length protein sequences were aligned by Clustalw and imported into the MEGA7 program. Phylogenetic analyses were conducted using the neighbor-joining (NJ) method.

### 2.6. Localization of MhAMT1;2 by GFP Fusion

The ORF of *MhAMT1;2* was subcloned into the expression vector pC1300 containing GFP gene, using the Kpn I and Xba I sites incorporated into the primers (Table S1). The recombinant plasmid was transformed into agrobacterium GV3101 by freeze-thaw method. Positive single colonies were picked and cultured in LB medium containing antibiotics at 28 °C until $OD_{600}$ was 0.8. After centrifugation, it was re-suspended in equal volume in the re-suspended solution containing 10 mM MES, 10 mM $MgCl_2$ and 200 μM AS. The suspension was injected into the lower epidermis of mature leaves of 5-week-old tobacco by syringe, and the tobacco leaves infected with Agrobacterium transformed with empty plasmid were used as control. After two days of normal culture, the leaves were observed by laser confocal microscope and photographed.

### 2.7. Functional Complementation of MhAMT1;2 in NH₄⁺ Uptake-Defective Yeast Mutant

The yeast strain 31019b (Δ*mep1*, Δ*mep2*, Δ*mep3*, *ura3*) was unable to grow on the medium containing less than 5 mM $NH_4^+$ as the sole N source [16]. The ORF of *MhAMT1;2* was subcloned into the expression vectors pYES2, and the pYES2-*MhAMT1;2* was transformed into 31019b by PEG/LiAc method and screened on the YNB medium containing 2 mM L-arginine. Positive single colonies were cultured in liquid YNB medium in an incubator shaker at 30 °C until $OD_{600}$ was 0.5–0.6. After centrifugation, the sediment was resuspended in sterile ultrapure water until the final $OD_{600}$ was 1.0, and it was continuously diluted 10 times to $10^{-1}$, $10^{-2}$, $10^{-3}$. For complementation tests, 6 μL aliquots of cell suspensions from each dilutes were spotted on YNB mediums containing 0.02, 0.2, 2 mM $NH_4Cl$. For mechanism tests, different concentrations of resuspension were spotted on YNB mediums containing DEA, CCCP and different pH. The 31019b strain transformed with an empty vector was used as a control and cultured in a constant temperature incubator at 30 °C for about 4 days, then photographed and recorded.

### 2.8. Functional Complementation of MhAMT1;2 in NH₄⁺ Uptake-Defective Arabidopsis Mutants

Transgenic Arabidopsis lines (*qko-MhAMT1;2*) were generated by expressing *MhAMT1;2* in the Arabidopsis quadruple insertion line *qko (atamt1;1, atamt1;2, atamt1;3, atamt2;1)* [13]. The ORF of *MhAMT1;2* was subcloned into the overexpression vector super1300. The

recombinant plasmid was transformed into *Agrobacterium* GV3101 by freeze-thaw method. Arabidopsis plants were transformed with each construct by the floral dipping method. Homozygous lines of the T2 and T3 generation were selected by segregation analysis for hygromycin resistance and used for subsequent phenotypic analysis.

Arabidopsis seeds were surface sterilized and plated onto a half-strength MS medium solidified with agar. After 1 week, the seedlings were transferred to the substrate and cultured under the following conditions: 12/12 h light/dark; light intensity 280 mmol m$^{-2}$ s$^{-1}$; temperature 22 °C; 70% humidity. After 5 weeks of culture, the plants were photographed and the aboveground fresh weight was weighed.

### 2.9. Statistical and Graphical Analyses

The data were presented as means ± standard errors of three biological replicate samples. All data were statistically analyzed using the SPSS 16.0 software by one-way ANOVA (SPSS, Chicago, IL, USA), followed by Duncan's multiple range tests. A *p*-value < 0.05 was identified as significant. Graphs were produced using Sigmaplot 10.0 software and arranged using Adobe Photoshop 7.0.

## 3. Results

### 3.1. Changes of Leaf Relative Water Content under Drought and Rewatering

Under drought treatment, the leaf relative water content of *M. hupehensis* plants decreased continuously from the first to seventh days of drought treatment (Figure 1). It decreased the fastest from the fifth to seventh days, and reached the lowest point 64% on the seventh day. After rewatering, the leaf relative water content increased rapidly to 81% on the eighth day.

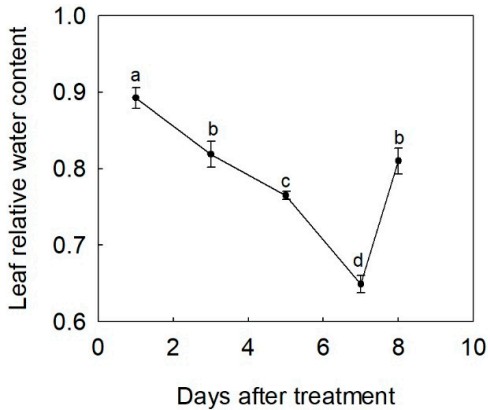

**Figure 1.** The leaf relative water content of *Malus hupehensis* plants under drought and rewatering. The plants were watered with saturated water one day in advance, then treated for 7 days without any water, and rewatered on the evening of 7th day. Bars labelled with different letters indicate significant difference.

### 3.2. Concentrations of NH$_4^+$ and NO$_3^-$ in Roots and Leaves

The concentrations of NH$_4^+$ and NO$_3^-$ in roots and leaves were measured to explore the effects of drought and rewatering on N uptake. The NO$_3^-$ content was significantly lower on the third, fifth, and seventh days than on the first day of drought treatment (Figure 2A). Even after rewatering, it did not return to the level of the first day. On the contrary, NO$_3^-$ content in the leaves showed an opposite trend to that in the roots. Compared with first day, the NO$_3^-$ content increased significantly on the third, fifth and seventh days, and declined to the level of the first day after rewatering (Figure 2B).

In addition, the root NH$_4^+$ content did not change significantly on the first and third days, then decreased significantly from the third to seventh days, and increased to the level of the first day after rewatering (Figure 2C). The content of NH$_4^+$ in leaves did not change significantly under drought treatment, but increased significantly after rewatering (Figure 2D).

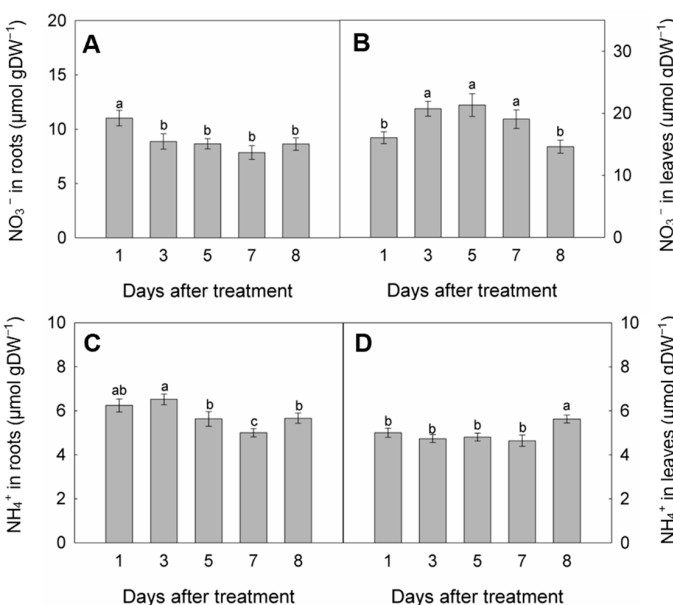

**Figure 2.** The effect of drought and rewatering on the content of $NO_3^-$ (**A,B**) and $NH_4^+$ (**C,D**) in leaves and roots of *Malus hupehensis* plants. Bars labelled with different letters indicate significant difference.

### 3.3. Activities Analysis of Key Enzymes Involved in N Assimilation

Under drought stress, the activities of enzymes related to N assimilation also changed in leaves and roots. Consistently in both roots and leaves, NR activity decreased notably during drought treatment and increased after rewatering. Root NR activity declined continuously in the first five days of drought treatment, remained stable on the seventh day, and increased slightly after rewatering (Figure 3A). The change of NR activity in leaves was relatively gentle, which decreased significantly in the third to fifth days of drought treatment, and returned to the level of the first day after rewatering (Figure 3B). GS activity showed the same trend in leaves and roots (Figure 3C,D). In the early stage of drought treatment, GS activity reduced rapidly, then increased and remained at a relative higher level throughout the subsequent treatment (Figure 3D).

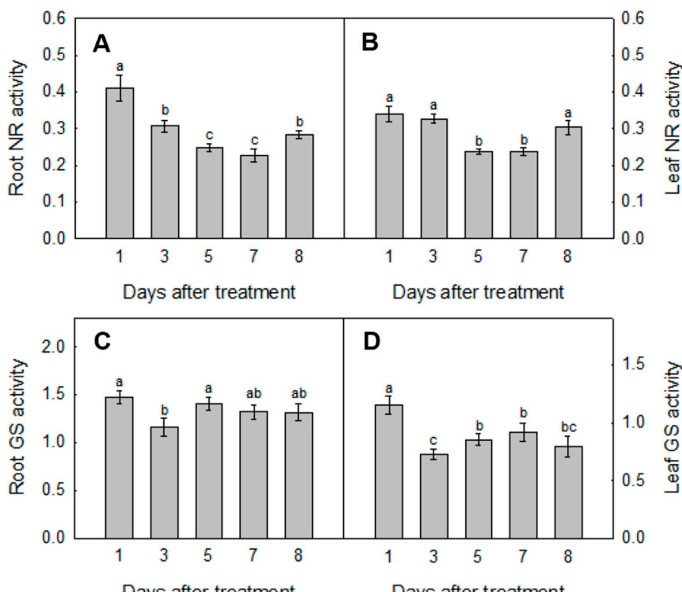

**Figure 3.** The effect of drought and rewatering on activities of nitrate reductase (NR; nmol $NO_3^-$ $h^{-1}$ $mg^{-1}$ protein) and glutamine synthetase (GS; $h^{-1}$ $mg^{-1}$ protein) in roots (**A,C**) and leaves (**B,D**) of *Malus hupehensis* plants. Bars labelled with different letters indicate significant difference.

### 3.4. Expression Changes of AMTs and NRTs under Drought and Rewatering

In order to find out the expression pattern of *AMTs* and *NRTs* during drought and rewatering, the transcript abundance of some key *AMTs* and *NRTs* were investigated. In roots, the transcriptional level of *AMT1;5* and *NRT2;4*, *AMT4;3* and *NRT2;5* showed similar trends (Figure 4A). With the deepening of drought stress, the transcription abundance of *AMT1;5/4;3* and *NRT2;4/2;5* continued to decrease, and reached the lowest point on the seventh day. After rewatering, *AMT1;5* and *NRT2;4* were up-regulated clearly on the eighth day, whereas expression levels of *AMT4;3* and *NRT2;5* were not obviously altered. Noteworthily, the expression level of *AMT1;2* in roots was particularly induced by drought. It increased continuously with the deepening of drought stress and reached the highest point on the seventh day, then decreased rapidly after rewatering.

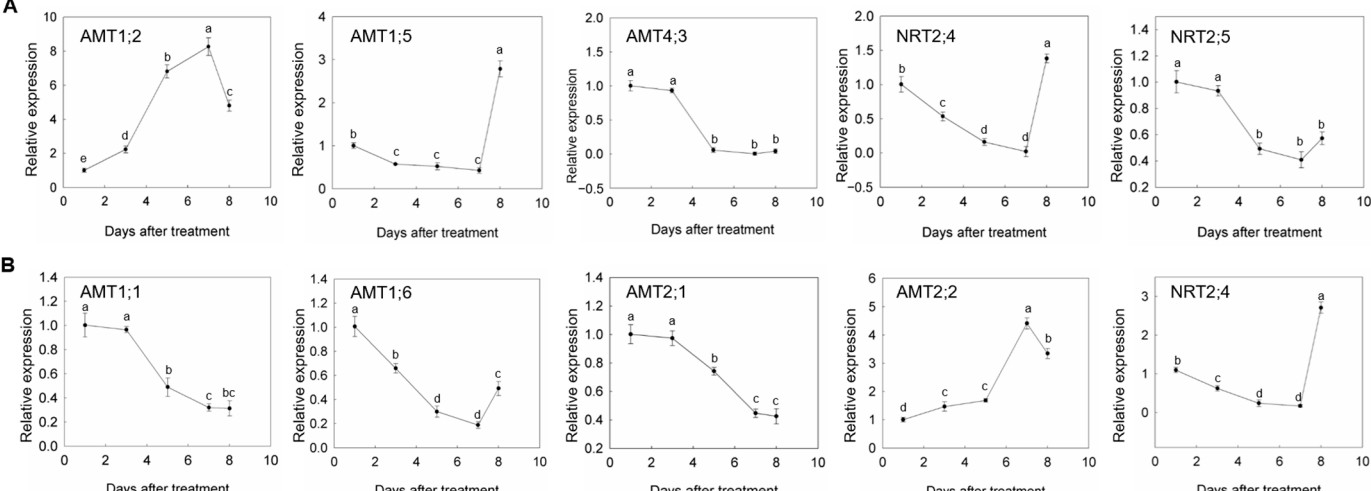

**Figure 4.** The effect of drought and rewatering on expression levels of key *AMTs* and *NRTs* in roots (**A**) and leaves (**B**) of *Malus hupehensis* plants. Bars labelled with different letters indicate significant difference.

The transcription level of *AMT1;6* and *NRT2;4* in leaves showed the same trend. Their transcription level continued to decrease under drought treatment, reached the lowest level on the seventh day and increased significantly after rewatering (Figure 4B). Conversely, the expression level of *AMT2;2* showed a completely opposite trend. The transcription level of *AMT1;1/2;1* also declined continuously with the deepening of drought, but it still showed a downward trend after rewatering on the eighth day. In addition, the expression levels of other investigated *AMTs* and *NRTs* did not show a clear trend with the change of soil water content (Figure S1).

### 3.5. Isolation of MhAMT1;2 Gene from Malus hupehensis

The function of *AMT1;2* was further explored as its expression level increased significantly in response to drought stress in *M. hupehensis* roots. We successfully cloned the full-length CDS sequence of *MhAMT1;2* from *M. hupehensis*. It was 1515 bp long and encoded a polypeptide of 505 amino acid residues. To further dissect the biological function of *MhAMT1;2*, we performed transmembrane (TM) prediction and phylogenetic analysis. MhAMT1;2 protein contained 11 TM domains, and the AMT-specific domain was found in TM5 (Figure 5A). Phylogenetic tree analysis implied that MhAMT1;2 was closely clustered with LjAMT1;2, LeAMT1;2 and AtAMT1;2, with a 79.65%, 77.22% and 77.91% amino acid identity, respectively (Figure 5B). Moreover, the tissue-specific expression analysis indicated that *MhAMT1;2* was mainly expressed in roots. It also showed a relatively low transcription level in mature leaves, and fairly low transcription levels in other investigated tissues, including stem, young leaves and senescing leaves (Figure 5C).

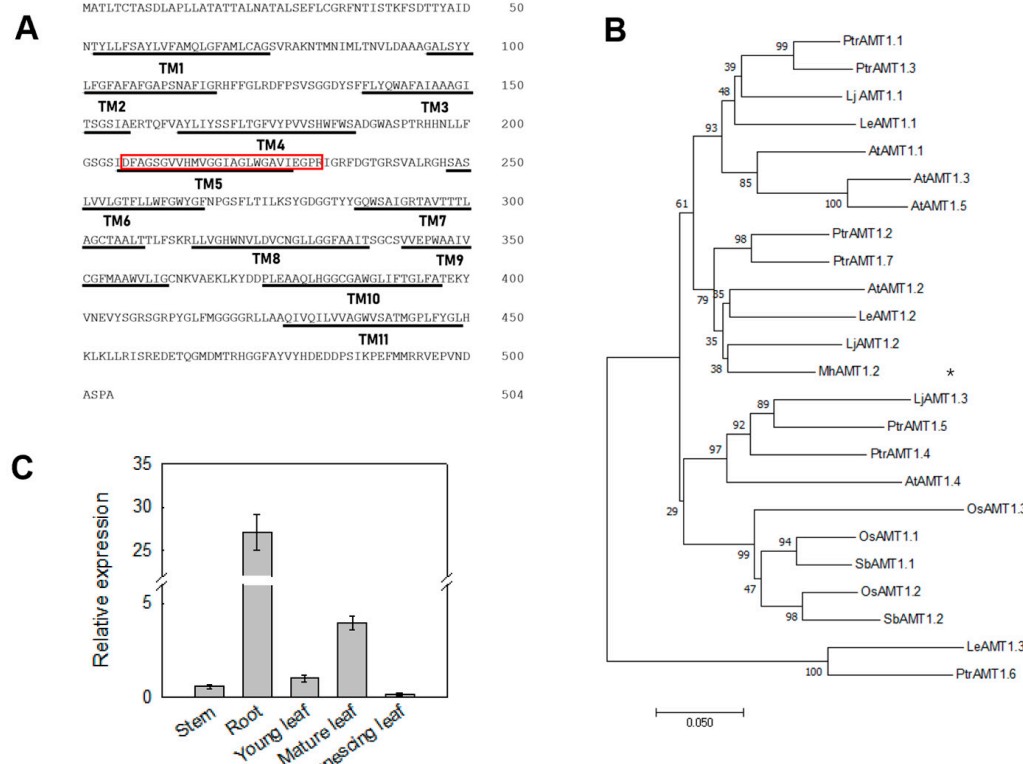

**Figure 5.** Phylogenetic analysis, protein sequence analysis of MhAMT1;2, and tissue-specific expression of its coding gene. (**A**) Predicted transmembrane (TM) domains in *MhAMT1;2*. TM domains were underlined. The AMT-specific domain was boxed in red. (**B**) phylogenetic tree construction of known AMT1 members in plants. *MhAMT1;2* was indicated with an asterisk. Scale bar stands for genetic distance. (**C**) Expression analysis of *MhAMT1;2* in different tissues of *Malus hupehensis* plants.

### 3.6. Localization of MhAMT1;2 in Tobacco Epidermal Cells

The results showed that the fluorescence signal appeared on the cell membrane, cytoplasm and nucleus in the tobacco cells expressing pC1300-GFP alone. The fluorescence signal of pC1300-MhAMT1:2-GFP fusion protein specifically appeared on the cell membrane (Figure 6), which indicated that MhAMT1;2 protein was localized in the cell membrane.

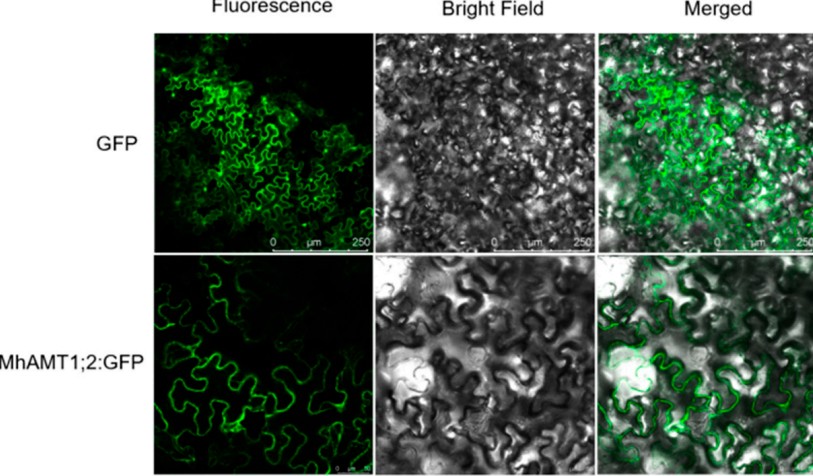

**Figure 6.** Subcellular localization of transiently expressed MhAMT1;2:GFP in tobacco leaf cells. GFP: pC1300-GFP; MhAMT1;2:GFP: pC1300-MhAMT1;2:GFP. Fluorescence: image of GFP expression under confocal microscopy; Bright-field: image of GFP expression under bright-field microscopy; Merged: overlaid image of corresponding bright-field and fluorescence images.

### 3.7. Transport Capacity of MhAMT1;2

To clarify the fundamental function of MhAMT1;2, the yeast mutant strain 31019b, which was defective in $NH_4^+$ uptake, was conducted to evaluate $NH_4^+$ uptake through MhAMT1;2. Figure 7 showed that 31019b cells expressing *MhAMT1;2* grew normally on the YNB medium with different concentrations of $NH_4^+$ as sole N source, and the yeast grew better as the $NH_4^+$ concentrations increased in the medium. Meanwhile, as a negative control, no growth was observed in cells transformed with the empty vector pYES2. These results indicated that MhAMT1;2 could mediate $NH_4^+$ absorption in yeast.

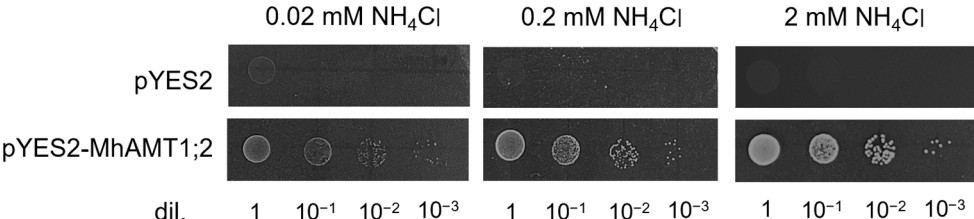

**Figure 7.** Functional analysis of MhAMT1;2-mediated $NH_4^+$ uptake in yeast. Yeast strains 31019 b transformed with pYES2 or pYES2-*MhAMT1;2* were grown on YNB medium, supplemented with different concentrations of $NH_4Cl$. Final dilutions are indicated by 1, $10^{-1}$, $10^{-2}$, and $10^{-3}$.

### 3.8. Proton and ATP Dependence of MhAMT1;2-Mediated Transport Process

We further analyzed the influencing factors of MhAMT1;2-mediated transport process, e.g., different pH and energy. Figure 8A showed that MhAMT1;2-expressing cells grew normally on YNB plate at pH = 4.8, 5.8, 6.8, whereas no growth was observed in yeast cells transformed with empty vector. Moreover, MhAMT1;2-expressing cells grew slightly better in an acidic environment (pH = 4.8), which indicated that the $NH_4^+$ transport ability of MhAMT1;2 could be affected by external protons.

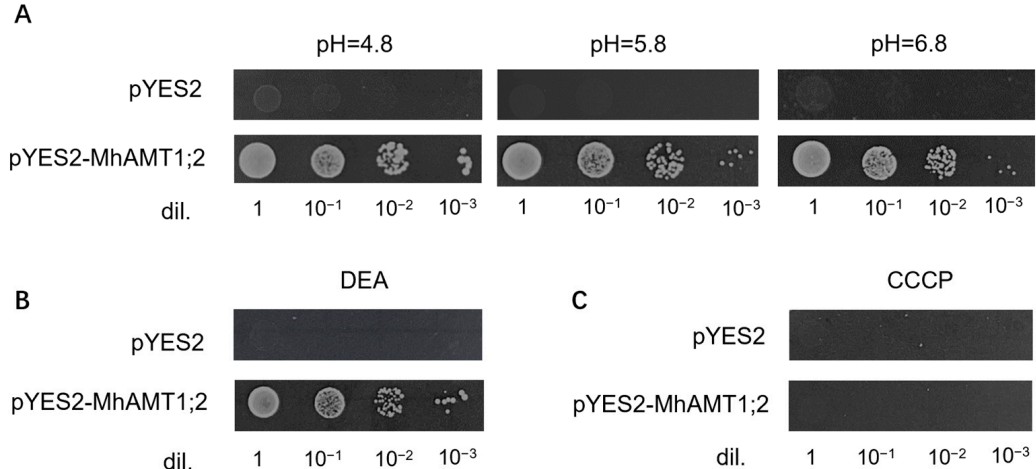

**Figure 8.** Proton and ATP dependence of $NH_4^+$ uptake by MhAMT1;2. (**A**) Effects of external pH on the growth complementation of yeast cells. (**B**) Effects of protonophore DEA on the growth complementation of yeast cells. (**C**) Effects of CCCP on the growth complementation of yeast cells. The cells were growing on YNB plates containing 2 mM $NH_4Cl$ as N source. DEA (100 μM) or CCCP (100 μM) was added to the medium prior to the solidification of the plates.

In Figure 8B, MhAMT1;2-expressing cells grew normally in the presence of protonophore DEA, indicating that the transport of $NH_4^+$ by MhAMT1;2 did not depend on the $H^+$ concentration gradient on both sides of cell membrane. Furthermore, the addition of CCCP, which is an uncoupling agent, resulted in no growth signs of yeast cells transformed with *MhAMT1;2* (Figure 8C), suggesting that $NH_4^+$ transport by MhAMT1;2 was an ATP-dependent active process.

### 3.9. Functional Complementation of MhAMT1;2 in NH₄⁺ Uptake-Defective Arabidopsis Mutants

Compared with wild-type Arabidopsis (Col-0), the growth of the quadruple insertion line *qko* was significantly inhibited, with a 48.3% decrease in shoot biomass (Figure 9). Notably, when *MhAMT1;2* was overexpressed in *qko*, the growth was partially restored and the shoot biomass reached 75.2% of Col-0, indicating that overexpression of *MhAMT1;2* could restore partial NH₄⁺ transport capacity of *qko*.

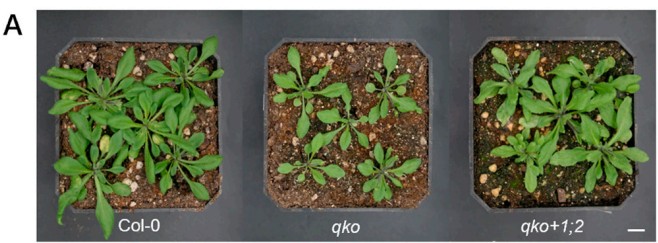
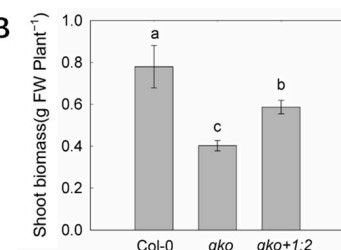

**Figure 9.** Growth of the wild type (Col-0), the quadruple insertion line *qko* and the transgenic line expressing *MhAMT1;2 (qko+1;2)*. (**A**) Growth of Col-0, *qko* and *qko+1;2* plants for six weeks. Bar = 1 cm. (**B**) The shoot fresh weight (FW) of Col-0, *qko* and *qko+1;2* plants in (**A**). Bars labelled with different letters indicate significant difference.

## 4. Discussion

### 4.1. Changes of N Uptake and Assimilation under Drought and Rewatering

The physiological activities of plants have shown to be greatly affected by soil moisture. Under drought treatment, the relative water content continued to decline in leaves of *M. hupehensis*, thereafter restored significantly after rewatering (Figure 1). Meanwhile, the internal content of NH₄⁺ and NO₃⁻ were also affected by drought stress and rewatering. In roots of *M. hupehensis*, both of the NH₄⁺ and NO₃⁻ concentrations decreased under drought and slightly recovered after rewatering (Figure 2A,C), which was in keeping with many previous studies [3,28]. However, the changes of NH₄⁺ and NO₃⁻ content in leaves were quite different. The leaf NH₄⁺ content was relatively stable during drought and increased significantly after rewatering, whereas the NO₃⁻ content was enhanced significantly under drought and thereafter reduced after rewatering (Figure 2B,D). Similar results have also been reported in sweet potato seedlings [6]. Previous research has shown that water deficit caused by drought could decrease NO₃⁻ uptake by influencing stomatal conductance and injuring membrane permeability and active transport [29]. Besides reducing root N uptake, drought stress also affected the distribution of N in the whole plant, resulting in a decrease in N allocation to roots, and an increase in soluble non-protein N in leaves due to protein degradation [3,30]. Moreover, drought-induced leaf NO₃⁻ accumulation may be related to its role in osmotic regulation of guard cells during stomatal closure, which was also reported in drought-stressed *M. prunifolia* seedlings [8].

After being absorbed by plant roots, NO₃⁻ can be reduced to nitrite by NR, thereafter form NH₄⁺ by nitrite reductase [31]. NH₄⁺ can be subsequently incorporated into glutamic acid by GS and glutamate synthase [32]. In our study, the activities of enzymes involved in N assimilation was also affected by drought and rewatering. NR has been reported as a sensitive enzyme to drought stress, and its activity consistently declined under drought and restored partially after rewatering in both roots and leaves (Figure 3A,B). Consistently results have also been reported in *Ipomoea batatas* and *P. simonii* [6,33]. In addition, the changes of GS activity under drought and rewatering were relatively stable. It decreased transitorily at the early stage of drought, then remained at a relatively higher level throughout the subsequent treatment (Figure 3C,D), which may be one of the reasons for the decrease of NH₄⁺ concentration in roots at the later stage of drought treatment. Furthermore, the enhanced NR activity and unchanged GS activity after rewatering may have partially contributed to the increase of NH₄⁺ concentration in roots of *M. hupehensis* plants.

Additionally, the transcript changes of *AMTs* and *NRTs* under drought stress have also been studied in many plants. In *P. simonii*, most of the predicted *NRT* genes and five *AMT* genes were down-regulated by drought stress [7,33]. The expression level of *SlAMT1-2* also showed a sharp decline in roots and leaves of *S. lycopersicum*. *TaAMT1;2* was induced by drought stress, while *TaAMT1;1* and *TaAMT2;1* was inhibited in roots of *T. aestivum* [21]. These results mentioned above indicated that members of plant *AMT* family showed various expression response to drought stress, suggesting that these genes may play distinct roles in varying plants. In the present study, the expression level of most investigated genes showed a downward trend with the duration of drought (Figure 4), which was consistent with the decrease of $NO_3^-$ and $NH_4^+$ content in the roots of *M. hupehensis*. Some genes were up-regulated in roots after rewatering, e.g., *AMT1;5* and *NRT2;4*, which may participate in the upswing of inorganic N absorption and contribute to the increment of inorganic N content after rewatering. It is worth noting that two genes were significantly up-regulated under drought and down-regulated after rewatering, including *AMT1;2* in roots and *AMT2;2* in leaves (Figure 4). We speculated that *AMT2;2* may play roles in maintaining $NH_4^+$ transport to leaves and *MhAMT1;2* may participate in stimulating $NH_4^+$ uptake by roots when *M. hupehensis* plants suffering drought stress. Therefore, functional study of *MhAMT1;2* in *M. hupehensis* was further performed.

### 4.2. MhAMT1;2 Is a ATP-Dependent High Affinity $NH_4^+$ Transporter

Plants take up and transport $NH_4^+$ mainly through members of AMT family [7,12]. Previous research has reported two plant AMT transport systems with different affinities, e.g., low affinity transport systems (LATs) and high affinity transport systems (HATs), in plant roots to adapt to changes in external N levels and environmental conditions [34]. After isolation and identification of the first plant $NH_4^+$ transporter in Arabidopsis by Ninnemann et al. [35], plant AMT gene family has been reported in several plant species [14,16]. This gene family can be mainly divided into AMT1 and AMT2 subfamilies [36]. The number of AMT1 subfamily members varies in different plant species: there are 5 AMT1 in Arabidopsis [12], only 3 in rice and crowtoe [14,16], and 7 in poplar [37]. Recently, 15 *MdAMTs* have been identified in apple genome, 8 of which belong to AMT1 subfamily and 7 belong to AMT2 subfamily [22]. In the present study, we successfully cloned the *MhAMT1;2* gene from *M. hupehensis*, with a length of 504 a.a. for encoded protein. Further analysis showed that protein sequence of MhAMT1;2 contained the complete conserved domain of plant AMT family (Figure 5A), which was consistent with some functional AMTs previously studied in other plant species [37,38].

Through phylogenetic tree construction of plant known AMT1 members, we observed that MhAMT1;2 was closely clustered with LjAMT1;2, LeAMT1;2 and AtAMT1;2 (Figure 5B). AtAMT1;2 is one of the most well-studied AMT transporters by now. It is located in root endothelial cells and is responsible for 18–26% of total $NH_4^+$ uptake in Arabidopsis roots [9,22]. Similar to AtAMT1;2, we found that MhAMT1;2 was also predominantly expressed in roots (Figure 5C), and located in the plasma membrane through subcellular localization in tobacco leaves (Figure 6). These results suggested that MhAMT1;2 may participate in root $NH_4^+$ absorption from soil solution. So far, several root-expressed functional AMTs have been studied, e.g., *OsAMT1;1*, *PutAMT1;1*, *ZmAMT1.1a* and *ZmAMT1.3*. Overexpression of these genes generally resulted in enhanced absorption of $NH_4^+$ and consequently increased biomass under limiting $NH_4^+$ supply [18,39,40]. Consistently, we found that the transgenic lines expressing *MhAMT1;2* grew better than *qko* mutant, and resulted a 45% increase of shoot biomass (Figure 9), further indicating that *MhAMT1;2* was able to mediate $NH_4^+$ uptake into roots.

Besides plant-level genetic manipulation, functional complementation in defective yeast was a way commonly used to explore the function of plant *AMTs*. Our results showed that MhAMT1;2 restored the $NH_4^+$ absorption capacity of yeast 31019b when different concentrations of $NH_4^+$ were supplied in the medium as sole N source (Figure 7). Considering that MhAMT1;2 had the ability to transport $NH_4^+$ across the membrane under low concentration of $NH_4^+$ (0.02 mM), it was also supposed to be a high affinity

$NH_4^+$ transporter. Moreover, increased transcript abundance of *MhAMT1;2* was found in the roots of *M.hupehensis* under N deficiency [23], which further confirmed that it acted as a high affinity transporter in $NH_4^+$ uptake. Previous studies on yeast heterologous expression have also confirmed that plant AMT1s encode $NH_4^+$ transporters with high affinity [6,9,19,22].

In addition, some plant AMTs have been verified to be proton-independent $NH_4^+$ transporters, as the absorption of $NH_4^+$ by these AMTs were not affected by external pH, such as LeAMT1;1, PutAMT1;1 and OsAMT1;1 [39,41,42]. Unlike these AMTs mentioned above, PvAMT1;1 was found as a $H^+/NH_4^+$ symporter, and the $NH_4^+$ absorption by PvAMT1;1 varied with different external pH [43]. Similarly, the yeast transformed with MhAMT1;2 grew slightly better in acidic environment (pH = 4.8) than neutral environment (pH = 5.8) (Figure 8A), indicating that $H^+$ might be involved in $NH_4^+$ transport by MhAMT1;2. Further quantitative comparison of yeast growth at different external pH is still needed. After eliminating the proton gradient on both sides of cell membrane through DEA addition in the medium, the yeast cells could still grow normally (Figure 8B). Therefore, it is speculated that the transport of $NH_4^+$ by MhAMT1;2 was a proton-independent process. Moreover, the fact that yeast cells did not grow after the addition of uncoupler CCCP indicated that MhAMT1;2 was an ATP-dependent $NH_4^+$ transporter.

Previous research also demonstrated that the transcript level of *AMTs* can be affected by external N status and abiotic stress. Some genes could be induced by N deprivation in roots, e.g., LeAMT1;1, LjAMT1;2 and PtrAMT1;2 [37,44,45], whereas some genes were downregulated. For example, the expression level of *ZmAMT1;1a* and *ZmAMT1;3* declined during N starvation periods [40]. Besides N regime, the expression level of several *AMTs* also altered in response to osmotic stress. Salt stress increased the transcript level of *PsAMT1.2* in *P. simonii* [19]. The expression level of *TaAMT1.2* was induced by drought stress while *TaAMT1.1* and *TaAMT2.1* was inhibited [21]. *PsAMT1;2* and *PsAMT1;6* were also upregulated in drought-stressed roots [7]. In this study, we also found that the expression level of *MhAMT1;2* raised significantly in roots under drought condition. However, the regulation mechanism of *AMTs* in response to various abiotic stress remained unknown, and the relationship between AMT-mediate $NH_4^+$ uptake and drought tolerance is still needed to be further studied.

## 5. Conclusions

In this study, the changes of N uptake and assimilation of *M. hupehensis* plants under drought and rewatering were investigated. The results showed that certain measurement index decreased under drought and some extent restored after rewatering, including leaf relative water content, root $NH_4^+$ and $NO_3^-$ content and NR activity. Most of investigated *NRTs* and *AMTs* were down-regulated while *MhAMT1;2* was significantly up-regulated in the drought-stressed roots. Furthermore, it was confirmed that *MhAMT1;2* was mainly expressed in roots and localized to plasma membrane and it may encode an ATP-dependent $NH_4^+$ transporter, which can mediate high affinity $NH_4^+$ uptake in roots of *M. hupehensis*. Further research is still needed to figure out the role of MhAMT1;2 in $NH_4^+$ uptake during plant adaption or resistance to drought stress.

**Supplementary Materials:** The following supporting information can be downloaded at: https://www.mdpi.com/article/10.3390/horticulturae9040434/s1, Table S1: Primer sequences used in this work; Figure S1: The effect of drought and rewatering on expression levels of *AMTs* and *NRTs* in root (A) and leaves (B) of *Malus hupehensis* plants.

**Author Contributions:** Conceptualization, J.L. and L.H.; data curation, J.L., M.Y.; formal analysis, H.L. and G.Y.; writing—original draft preparation, J.L.; writing—review and editing, L.H. and Y.H. All authors have read and agreed to the published version of the manuscript.

**Funding:** This work was supported by the Scientific and Technological Innovation Programs of Higher Education Institutions in Shanxi (No. 2019L0358); the Scientific and Technological Research and Extension Program of Water Conservancy in Shanxi Province (NO. 2022GM045); the PhD Start-up Fund of Shanxi Province (No. SXYBKY2018014) and the Technology Innovation Fund of Shanxi Agricultural University (No. 2018YJ03).

**Data Availability Statement:** The data presented in this study are available upon request from the corresponding author.

**Acknowledgments:** We thank Yuan Lixing (College of Resources and Environmental Sciences, China Agricultural University) for kindly providing the ammonium uptake-deficient yeast strain 31019b and the Arabidopsis quadruple insertion line *qko*.

**Conflicts of Interest:** The authors declare no conflict of interest.

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
