# Peer review of "Functional Characterization of Ammonium Transporter MhAMT1;2 in Malus hupehensis"

_horticulturae, doi:10.3390/horticulturae9040434_

Round 1

Reviewer 1 Report

Dear the editor,

The manuscript entitled “Functional characterization of ammonium transporter MhAMT1;2 in Malus hupehensis” describes the drought responses in M. hupehensis and the characterization of MhAMT1;2 encoding an ammonium transport. Because relationship between the nitorgen utilization and the water status is an interesting subject in plant sciences, this research has potential to be valuable for readers in the related field.

As the manuscript focuses on the plant drought response, I would like to make a suggestion about the drought experiments first.

In the manuscript, the Malus plants have been subjected to a drought condition without supplying water for 7 days. From the manuscript, it is difficult to evaluate how the experiment is effective for studying the drought response in M. hupehensis. In other words, since many researchers would like to compare the results on the drought experiments, the authors should provide data that allow the researchers to compare the results and learn from the results provided in the manuscript. One possibility may be measuring water potential as an indicator in M. hupehensis in the experiment.

In addition, data on the concentration of ammonium and nitrate showed in Figure 2 are not sufficient for concluding the status of nitrogen uptake since the concentration is a result not only of the uptake from the soil/water, but also the reduction and transport within the plants.

Another suggestion is on the experiments of MhAMT1;2. The purpose of the characterization of MhAMT1;2 is not clear. In other words, I hardly understand what of the aspects on ammonium transporter the authors aimed to reveal in this work. Since the authors have been identified 15 AMT members, I am wondering if obtained results would be different from the data in the manuscript when the authors would use another AMT gene in the experiments of cellular localization, yeast complementation and overexpression in Arabidopsis.

Minor points.

1.     Line 12, please check if it is o.k. to use “NR” without full description.

2.     Line 52, “knockout of OsAMT1;1-1;3” is not clear statement. It would be the triple mutants of OsAMT1;1, OsAMT1;2 and OsAMT1;3.

3.     Line 56, on PutAMT1, please specify the species, Puccinelia tenuiflora.

4.     Line 83, for tissue specific expression analysis, what plants are used and what was the growth condition for the sampled plants.

5.     Figure1, please explain the alphabets (a-c) in the figure legend. Please add an alphabet on 7th day.

6.     Line 208, “un-regulated” seems to be typo.

7.     Related to Figure 4, add statistical analyses on the expression data.

8.     Related to Figure 7 and Figure 8, please add quantification data. Showing the pictures of the yeast growth is not scientifically acceptable.

Reviewer 2 Report

Dear appreciated authors,

The manuscript “Functional characterization of ammonium transporter 2 MhAMT1;2 in Malus hupehensis” was found interesting, new and appears to be scientifically sound. Author's paper should be accepted for publication, after minor revisions, because the paper represents the contribution for the science.

 However, there are a few details which should be considered (see below) and some revisions have to be made.

Line 413 - 422: In this study, the changes of N uptake and assimilation of M. hupehensis plants under drought and rewatering were investigated. The results showed that certain measurement index decreased under drought and some extent restored after rewatering, including leaf relative water content, root NH4+ and NO3 - content and NR activity. Most of investigated NRTs and AMTs were down-regulated while MhAMT1;2 was significantly up-regulated in the drought-stressed roots. Furthermore, it was confirmed that MhAMT1;2 was mainly expressed in roots and localized to plasma membrane and it may encoded a ATP-dependent NH4+ transporter, which can mediate high affinity NH4+ uptake in roots of M. hupehensis. Further research is still needed to figure out the role of MhAMT1;2 in NH4+ uptake during plant adaption or resistance to drought stress.

Best regards,

NL

Round 2

Reviewer 1 Report

Dear the editor,

Most of my pointed issues have been rewritten in the revised manuscript.

I still have a concern related to Figure 8. I do understand the yeast complementation test, but the interpretation of the comparison of yeast growth among different pH media should be more carefully. Without quantitative comparison, it is not simple to conclude the growth difference among the different pH media. It would be better to weaken the statement about the results of the experiment in the manuscript.

I found, in the discussion section, some proofreading is required in the revised manuscript.

Best
